# Sustained Effects of High-Intensity Interval Exercise and Moderate-Intensity Continuous Exercise on Inhibitory Control

**DOI:** 10.3390/ijerph18052687

**Published:** 2021-03-07

**Authors:** Shudong Tian, Hong Mou, Fanghui Qiu

**Affiliations:** School of Physical Education, Qingdao University, Qingdao 266071, China; tsdtyxy@163.com (S.T.); mouhong2021@126.com (H.M.)

**Keywords:** HIIE, MICE, inhibitory control, time course

## Abstract

This study examined the immediate and sustained effects of high-intensity interval exercise (HIIE) and moderate-intensity continuous exercise (MICE) bouts on inhibitory control in young adults. Participants (*n* = 41) engaged in (1) a session of HIIE, involving 10 one-minute runs on a treadmill at an intensity targeting 85–90% HR_max_ interspersed with self-paced walking at 60% HR_max_; (2) a session of MICE, involving a 20 min run on a treadmill at an intensity of 60–70% HR_max_; and (3) a control session, involving 24 min of resting on separate days in a counterbalanced order. Using a flanker task, inhibitory control was assessed before the intervention (t_0_), immediately after the session (t_1_), and then at 30 min (t_2_), 60 min (t_3_), and 90 min (t_4_) after the session. During the flanker task, the response time (RT) for incongruent trials immediately after HIIE was significantly shortened compared to that before exercise. This shortened RT was sustained for 90 min post-exercise during recovery from HIIE. Interference scores of RT were also reduced after HIIE, benefitting inhibitory control, and were maintained for 90 min post-exercise. Reduced accuracy interference scores were recorded following HIIE compared to the control session. Improvements in inhibitory control elicited by HIIE were sustained for at least 90 min post-exercise. In contrast, an improvement in inhibitory control was not observed during the MICE session. HIIE might represent a time-efficient approach for enhancing inhibitory control.

## 1. Introduction

Inhibitory control refers to the ability to resist internal impulses or external lures, and it inhibits inappropriate or prepotent responses. Inhibitory control is necessary for the regulation of behavior and emotions, and is associated with success in daily life [1]. A growing body of research has demonstrated that acute exercise has transient effects on inhibitory control [2,3,4,5,6,7]; however, the results depend on the characteristics of the exercise intervention, including the intensity and protocols of acute exercise bouts [2]. The influence of continuous exercise, as a traditional form of exercise, on inhibitory control has been widely investigated [8,9,10,11]. Most studies have documented improved inhibitory control after a bout of moderate-intensity continuous exercise (MICE) [11,12,13,14], whereas other studies demonstrated that inhibitory control did not change after MICE [15,16,17]. High-intensity interval exercise (HIIE) is a novel and time-efficient physical activity [18], which is now acknowledged as a key approach for cognitive and mental health [19]. Evidence from experimental studies has shown that HIIE elicits greater benefits for inhibitory control compared to MICE [20,21].

Various studies have demonstrated that improvement in cognitive performance by acute exercise can be sustained for a certain period of time. Experimental studies have shown that the benefits of acute exercise can last up to 45 min after exercise [22,23]. However, the time course of changes in inhibitory control function might depend on the protocol of the preceding exercise session. Tsukamoto et al. [20] have demonstrated that HIIE and MICE protocols can improve inhibitory control immediately after exercise. However, the improvement in inhibitory control was maintained for 30 min after the HIIE session, during which the performance after MICE returned to pre-exercise levels. Various studies have shown that intermittent exercise significantly improves performance in the flanker task for over 60 min [24], whereas improved inhibitory control only lasts up to 30 min after continuous exercise [25]. One meta-analysis has indicated that when the cognitive test (including inhibitory control) is administered following a delay after exercise, more intense exercise produces the strongest effects; however, when the cognitive performance test is administered immediately after exercise, lighter-intensity exercise produces greater benefits [26]. Therefore, the effect of different exercise protocols on sustainably improving inhibitory control should be examined.

The current study examined the immediate and sustained effects of HIIE and MICE on inhibitory control in young adults. Using a within-subjects design, 41 participants engaged in one HIIE session, one MICE session, and one control session on separate days in a counterbalanced order. The flanker task was performed to assess inhibitory control before the intervention (t_0_) and at four time points after the intervention, including immediately (t_1_), 30 min (t_2_), 60 min (t_3_), and 90 min (t_4_). Because previous studies have demonstrated that different exercise intensities and modalities are potential moderators of exercise-induced inhibitory control [20,26], we hypothesized that (1) HIIE would elicit a more positive improvement in inhibitory control compared to MICE and control sessions, and (2) improvement in inhibitory control would last longer after HIIE compared to after MICE.

## 2. Materials and Methods

### 2.1. Participants

Forty-one healthy participants (21 males and 20 females) from Qingdao University, China, participated in this study. All participants were right-handed, free of any reported neurological, cardiovascular, or pulmonary diseases, and had normal or corrected-to-normal vision. Participants were asked to refrain from any moderate-to-vigorous physical exercise 24 h before the experiments and to refrain from caffeine and alcohol 12 h before the experiments. The study protocol was approved by the local ethics committee of Qingdao University. All participants provided written informed consent before the experiment. Demographic characteristics and fitness data for all participants are provided in Table 1.

### 2.2. Procedure

The study was conducted using a within-subjects, repeated-measures design. It included one HIIE session, one MICE session, and one control session conducted in a counterbalanced order. These three sessions were separated by at least one week and were completed at approximately the same time of the day. Before the experiment, participants completed the Physical Activity Readiness Questionnaire (PAR-Q) [27], which identifies potential risk factors. Participants were asked to complete the flanker task before the intervention (t_0_) and at four time points after the intervention, including immediately (t_1_), 30 min (t_2_), 60 min (t_3_), and 90 min (t_4_).

### 2.3. Exercise Protocols

During the HIIE session, participants completed a 2 min warm-up at a self-determined speed, followed by 10 bouts of repeated 1 min runs on a treadmill at an intensity targeting 85–90% HR_max_ (208 - 0.7 × age) [28], interspersed with 1 min of self-paced walking [20,29]. After the last walking interval period, a 2 min walk was provided as a cool-down period. During the MICE session, participants completed a 2 min warm-up, followed by 20 min of running on a treadmill at an intensity of 60–70% HR_max_ [30], which was ended with a 2 min cool-down period. During the control session, participants sat quietly on a chair and read a book for 24 min. Before each session, each participant was fitted with a Polar H10 heart rate strap (Polar, Kemple, Finland), which was kept fitted until the end of each session. Ratings of perceived exertion (RPEs) were assessed at the end of HIIE and MICE interventions [31]. The HIIE, MICE, and control protocols are illustrated in Figure 1.

### 2.4. Flanker Task

To assess inhibitory control, a computer-based version of the flanker task [32] was administered with E-Prime 2.0 (Psychology Software Tools Inc., Pittsburgh, PA, USA). This task is very sensitive to acute exercise [33,34]. Visual stimuli included five black arrows presented on a white background in either a congruent (arrows facing the same direction, i.e., “> > > > >” or “< < < < <”) or an incongruent sequence (central arrow facing in different directions, i.e., “> > < > >” or “< < > < <”). Each experimental trial was initiated with a 300–500 ms presentation of a black fixation “plus” symbol on the white computer screen. Participants were instructed to respond as quickly and accurately as possible to the direction of the centrally presented arrow by pressing a button using their left index finger when the target arrow faced to the left (i.e., “<”) and pressing a button with their right index finger when the target arrow faced to the right (i.e., “>”). The task consisted of two blocks of 60 trials, with 30 congruent trials and 30 incongruent trials being presented in a randomized order in each block. Ten practice trials of the task were provided before the start of the first formal experiment. Subsequently, a stimulus was presented for 100 ms, followed by a 1100–1500 ms interval. Response time (RT) and accuracy for the flanker task were collected. The mean RT from response-correct trials and accuracy were calculated for each trial type (congruent and incongruent). The incongruent condition, relative to the congruent condition, necessitated the concurrent activation of both the correct response (elicited by the target) and the incorrect response (elicited by the flanking stimuli) before stimulus evaluation was completed; thus, a greater amount of interference control was required to inhibit the flanking stimuli and execute the correct response [35]. Interference score measures were obtained by simple subtractions across task conditions, providing information on changes in the speed and accuracy of information processing between congruent and incongruent trials [36].

### 2.5. Statistical Analysis

For the RT analysis, incorrect trials were first removed and then an outlier correction was performed by separately excluding trials with an RT of 3 standard deviations from the mean for each flanker condition (congruent and incongruent). Accuracies and RTs were analyzed using a three-way repeated-measures analysis of variance (ANOVA), with session (HIIE, MICE, and control), time point (t_0_, t_1_, t_2_, t_3_, and t_4_), and congruency (congruent and incongruent) being used as within-subject factors. Interference scores were examined separately for accuracy (congruent‒incongruent) and RT (incongruent‒congruent) using a 3 (session: HIIE, MICE, and control) × 5 (time: t_0_, t_1_, t_2_, t_3_, and t_4_) repeated-measures ANOVA. Mauchly’s test was used to examine spherical data, and the Greenhouse–Geisser correction was used to analyze non-spherical data. The Shapiro–Wilk normality test was applied to confirm normal distribution of data before the ANOVA. Paired *t*-tests with Bonferroni corrections for multiple comparisons were used for post hoc analysis if the ANOVA showed significant interactions between different factors. *p*-Values < 0.05 were considered significant. All statistical analyses were completed using the Statistical Package for Social Sciences software (SPSS version 25.0, Chicago, USA).

Sample size was calculated via the statistical power calculation (G*power 3.1.9.2) on a medium effect size (*η*^2^ = 0.17) [3], assuming α level of 0.05 and a desired power (1 - β) of 0.80 at the group level. Thus, corresponding to a required sample size of 29 subjects, we aimed to enroll more participants in the study to compensate for potential withdrawal. Ultimately, 41 participants volunteered to participate in the study.

## 3. Results

### 3.1. Accuracy

The three-way ANOVA showed a significant main effect of session on accuracy (*F*(2, 39) = 5.02, *p* = 0.012, *η*^2^ = 0.21). Post hoc analysis showed that accuracy was significantly higher in the HIIE session (94.52 ± 6.36) compared to the control session (92.94 ± 7.21, *p* = 0.009), while accuracy in the MICE session (94.17 ± 6.12) did not differ from the other two sessions (*p* > 0.1). A significant main effect of time point was detected (*F*(4, 37) = 3.43, *p* = 0.017, *η*^2^ = 0.27). Post hoc tests showed that accuracy was significantly higher 30 min after the intervention (t_2_) compared to before the intervention (t_0_) (*p* = 0.009). There was a significant main effect of congruency (*F*(1, 40) = 71.91, *p* < 0.001, *η*^2^ = 0.64), with higher accuracy in the congruent trials compared to the incongruent trials.

There was a significant interaction between session and congruency (*F*(2, 39) = 6.69, *p* = 0.003, *η^2^* = 0.26). Post hoc analysis showed that accuracy was significantly higher in the incongruent trials of the HIIE session (90.88 ± 9.11) compared to the control session (87.76 ± 11.18, *p* = 0.002), while accuracy in the MICE session (89.93 ± 8.71) did not differ from the other two sessions (*p* > 0.08 for both). There were no significant effects in the interactions of time point × congruency (*F*(4, 37) = 2.54, *p* = 0.056, *η*^2^ = 0.12), session × time point (*F*(8, 33) = 0.57, *p* = 0.792, *η*^2^ = 0.12), or session × time point × congruency (*F*(8, 33) = 0.21, *p* > 0.9, *η*^2^ = 0.05) (Figure 2a,b).

Analysis of the accuracy interference scores showed that there was a significant main effect of session (*F*(2, 39) = 6.69, *p* = 0.003, *η*^2^ = 0.26). Post hoc analysis showed that the accuracy interference score was significantly lower in the HIIE session (7.28 ± 9.07) compared to the control session (10.37 ± 10.40, *p* = 0.002), while the accuracy interference score in the MICE session (8.47 ± 8.16) did not differ from the other two sessions (*p* > 0.1 for both). There was no significant main effect of time point (*F*(4, 37) = 2.54, *p* = 0.056, *η^2^* = 0.22), and no significant effect in the interaction of session × time point (*F*(8, 33) = 0.21, *p* > 0.9, *η^2^* = 0.05) (Figure 2c). Mean accuracies and RTs of congruent and incongruent trials and interference scores in HIIE, MICE, and control sessions are shown in Table 2.

### 3.2. RT

The three-way ANOVA on RT revealed a significant main effect of time point (*F*(4, 37) = 2.85, *p* = 0.037, *η*^2^ = 0.24). The post hoc test showed that RT was significantly lower before intervention (t_0_) (420.99 ± 36.00) compared to 60 min (t_3_) (413.87 ± 34.21, *p* = 0.029) and 90 min (t_4_) (412.32 ± 33.00, *p* = 0.019) after intervention. A significant main effect of congruency was documented (*F*(1, 40) = 720.17, *p* < 0.001, *η*^2^ = 0.95), with longer RTs in the incongruent trials (448.61 ± 37.65) compared to the congruent trials (382.61 ± 32.72).

There was a significant interaction between time point and congruency (*F*(4, 37) = 5.65, *p* = 0.001, *η*^2^ = 0.38). The post hoc test showed that, for incongruent trials, RT was significantly slower before intervention (t_0_) compared to immediately after intervention (t_1_) (*p* = 0.02) and at 30 min (t_2_) (*p* = 0.004), 60 min (t_3_) (*p* = 0.006), and 90 min (t_4_) (*p* = 0.003) after intervention. No significant differences were observed between the time points for congruent trials (*p* > 0.5 for all). There was a significant interaction effect for session × time point × congruency (*F*(8, 33) = 2.51, *p* = 0.03, *η*^2^ = 0.38). Contrasts in the interactions between session, time point, and congruency revealed that RT was significantly slower in incongruent trials before the HIIE intervention (t_0_) compared to immediately after the HIIE intervention (t_1_) (*p* = 0.002) and at 30 min (t_2_) (*p* = 0.018), 60 min (t_3_) (*p* = 0.003), and 90 min (t_4_) (*p* = 0.045) after the HIIE intervention. No significant differences were detected between time points for incongruent trials in the MICE and control sessions (*p* > 0.063 for all). For the RTs of congruent trials, no significant differences were detected between the time points in any of the sessions (*p* > 0.085 for all). There was no significant main effect for session (*F*(2, 39) = 1.76, *p* = 0.186, *η*^2^ = 0.08) or significant effects in the interactions of session × congruency (*F*(2, 39) = 2.69 *p* = 0.081, *η^2^* = 0.12) or session × time point (*F*(8, 33) = 1.24, *p* = 0.309, *η*^2^ = 0.23) (Figure 3a,b).

Analysis of the RT interference scores showed a significant main effect of time point (*F*(4, 37) = 5.65, *p* = 0.001, *η*^2^ = 0.38). The post hoc test showed that the RT interference score was significantly higher before the intervention (t_0_) compared to immediately after the intervention (t_1_) (*p* = 0.001) and at 30 min (t_2_) (*p* = 0.002), 60 min (t_3_) (*p* = 0.046), and 90 min (t_4_) (*p* = 0.008) after the intervention. There was a significant interaction between session and time point (*F*(8, 33) = 2.89, *p* = 0.03, *η*^2^ = 0.38). The interaction contrasts revealed that RT interference score was significantly lower immediately after the HIIE intervention (t_1_) compared to before the HIIE intervention (t_0_) (*p* < 0.001). This shortened RT was sustained for 90 min (*p* < 0.04 for all). No significant differences were detected in the RT interference scores between the time points in the MICE and control sessions (*p* > 0.164 for all). There was no significant main effect of session (*F*(2, 39) = 2.69, *p* = 0.081, *η*^2^ = 0.12) (Figure 3c).

## 4. Discussion

This study assessed the sustainability of the effect of acute treadmill-based HIIE and MICE on inhibitory control in young adults. After the HIIE intervention, participants performed better in an inhibitory-control task, with performance being sustained for at least 90 min after the intervention. This improvement in inhibitory control was not detected in the MICE session. Thus, HIIE could represent an effective strategy for improving inhibitory control.

As a stressor, exercise can activate the autonomic nervous system and promote physiological and psychological arousal [37]. Physiological changes such as increased brain metabolism and increased oxygen and blood flow to the brain during exercise [38] could optimize the allocation of cognitive resources, promoting more efficient cognitive processing [39]. This study demonstrated that participants only performed better in the inhibitory-control task after the HIIE intervention, compared to the control session (resting). After HIIE, accuracy increased and RT selectively decreased in incongruent trials, and accuracy interference scores and RT interference scores decreased. These findings are similar to previous studies, showing that the HIIE intervention generated greater benefits for inhibitory control in young adults compared to MICE [20,21]. Using a modified flanker task, Kao et al. [21] showed that the RT of inhibitory control generally declined after MICE (20 min acute activity) and HIIE (9 min acute activity), compared with resting; however, only HIIE improved accuracy when the demand of the inhibitory control task increased. Thus, HIIE represents an effective intervention strategy for improving inhibitory control [40].

The present study failed to show an enhancement of inhibitory control after the MICE session compared to the control session, which was inconsistent with studies reporting improvements in inhibitory control after MICE interventions [6,41]. First, these discrepancies may stem from differences in the tasks assigned to participants. Several previous studies employing a modified color-word Stroop task showed an improvement of inhibitory control in young adults after a MICE intervention [6,20,41]. However, along with the present study, studies using a modified flanker task reported no significant effect of MICE on inhibitory control [42,43]. Second, the effect of MICE on inhibitory control might be influenced by differences in cardiorespiratory fitness levels. In the current study, MICE intensity was approximately 70% of HR_max,_ with a mean RPE of 10.85; however, moderate-intensity exercise represents an RPE of 12–13 according to the American College of Sports Medicine (ACSM) guidelines [44]. This phenomenon might be attributed to higher levels of cardiorespiratory fitness that produced lower levels of fatigue at the same target heart rate. Drollette et al. [45] showed that MICE significantly improved P300 amplitudes of correct trials in the flanker task in individuals with lower fitness levels; however, this effect was not observed in individuals with higher fitness levels. Third, it is also possible that the effect of one session of MICE may not be as apparent in changes in behavioral measures compared to changes in electrophysiological measures. In this regard, two studies have shown that electrophysiological changes have been confirmed after MICE without manifesting in significant behavioral measures of performance [33,46].

The present study demonstrated that inhibitory control improved immediately after HIIE and was sustained for at least 90 min. This finding supports several recent studies reporting that cognitive improvement could be maintained for a period of time after exercise [20,23,24]. For example, Tsukamoto et al. [20] showed that improved inhibitory control due to HIIE could be sustained for 30 min during recovery post-exercise. Two recent studies showed that improvement of inhibitory control could be maintained for 45 min after HIIE [22,23]. One meta-analysis showed that when cognitive function (including inhibitory control) was tested at a time point following exercise, more intense exercise produced the strongest effects [26]. Our findings validate previous studies reporting that HIIE improves inhibitory control [20,21], confirming that improvement in inhibitory control due to HIIE can be maintained for up to 90 min after exercise.

Exercise-induced enhancement of inhibitory control is associated with brain oxygenation [11]. Participation in acute exercise can help increase oxygen-rich blood flow to the brain [47], particularly in areas supporting inhibitory control (mainly the prefrontal cortex) [48]. However, Lambrick et al. [6] confirmed that blood flow and oxygenation in the prefrontal cortex increased after both intermittent exercise and MICE, but that inhibitory control was improved only after intermittent exercise. Thus, increased blood flow and oxygenation in the prefrontal cortex might explain the additional benefit of HIIE for inhibitory control, compared to MICE. Furthermore, Mcmorris et al. [49] suggested that improved performance in inhibitory control after HIIE is related to the exercise-induced release of catecholamines (e.g., dopamine and norepinephrine) in the brain (catecholamine hypothesis of the exercise–cognitive relationship). Catecholamines enhance cognitive performance and facilitate arousal [50]. Increased catecholamine release through intense exercise improves neural activity and central executive performance [49]. High-intensity exercise also reduces glucose uptake in the brain. Consequently, exercise could impact lactate metabolism in the brain by increasing the concentration of serum lactate, particularly in the prefrontal cortex regions [51]. Blood lactate is an important source of energy for the brain and is due to the compensatory action of reduced glucose uptake [52]. Because HIIE produces more lactate, it is more likely to improve inhibitory control. Tsukamoto et al. [20] reported that compared to MICE, HIIE generates significantly higher lactate levels at 30 min after recovering from exercise. Enhanced inhibitory control and exercise-induced increases in serum lactate concentrations might be linked. Thus, exercise-induced greater or more sustained increases in these biomarkers are more likely after HIIE compared to MICE.

The results of the present study have several limitations. First, cardiorespiratory fitness can modulate the effect of acute exercise on cognitive performance [2,26]. However, baseline measurements of cardiorespiratory fitness were not collected as part of this study. Thus, further studies are needed to test whether the differential effects of acute HIIT and MICE on inhibitory control are modulated by cardiorespiratory fitness. Second, the RPE in the current study was only 13.41 and 10.85 for HIIE and MICE, respectively. ACSM guidelines [44] state that the RPE of high- and moderate-intensity exercise should fall within 14–17 and 12–13 RPE, respectively. These slight differences might be explained by the fact that the current study assessed RPE immediately after exercise, whereas RPE has previously been assessed every 2 min during exercise [21]. Furthermore, the HR_max_ in this study was estimated from an equation (208 - 0.7 × age) [28], which might have biased the assessment of exercise intensity. This approach might have also contributed to the lower RPE under the HIIE and MICE conditions in this study compared to ACSM guidelines. In future studies, we will perform a maximal exercise test as a first trial to measure maximum heart rate with the aim of prescribing a more accurate evaluation of intensity, in parallel to assessing certain indicators, such as blood lactate.

## 5. Conclusions

The present study demonstrated how treadmill-based acute HIIE and MICE impact the sustainability of inhibitory control in young adults. Specifically, HIIE enhanced inhibitory control for up to 90 min following the cessation of exercise. However, this improvement in inhibitory control was not detected in the MICE session. Thus, HIIE represents a time-efficient approach for enhancing cognitive performance, with it being recommended that the public be made aware of this. For example, acute HIIE would be recommended for students before implementing a study, to achieve higher levels of concentration and efficiency. This temporary change in executive control is suggested to have an impact on academic performance and success in daily life [2,53].

## Figures and Tables

**Figure 1 ijerph-18-02687-f001:**
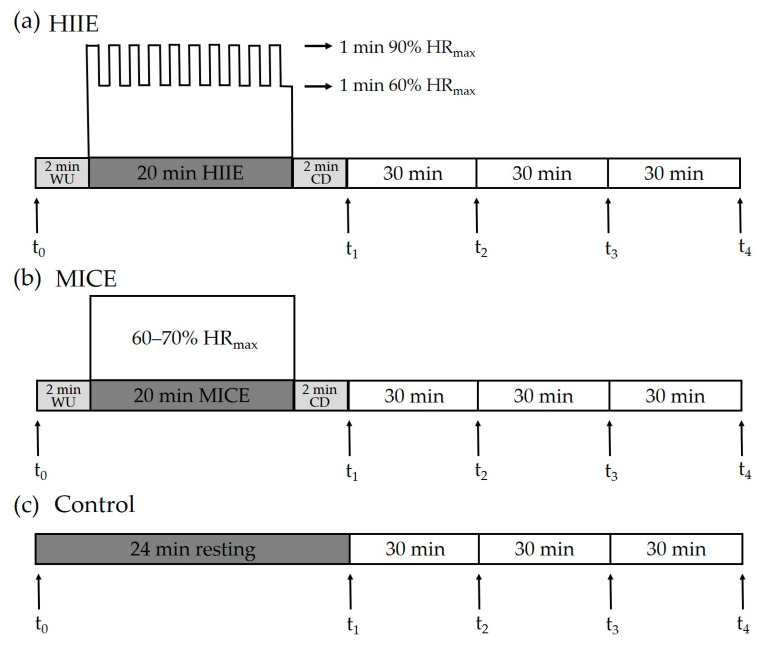
Experimental protocol. Inhibitory control was assessed pre-exercise/at rest (t_0_) and at four time points after intervention, including immediately (t_1_), 30 min (t_2_), 60 min (t_3_), and 90 min (t_4_). HIIE: high-intensity interval exercise. MICE: moderate-intensity continuous exercise. WU: warm-up. CD: cool-down.

**Figure 2 ijerph-18-02687-f002:**
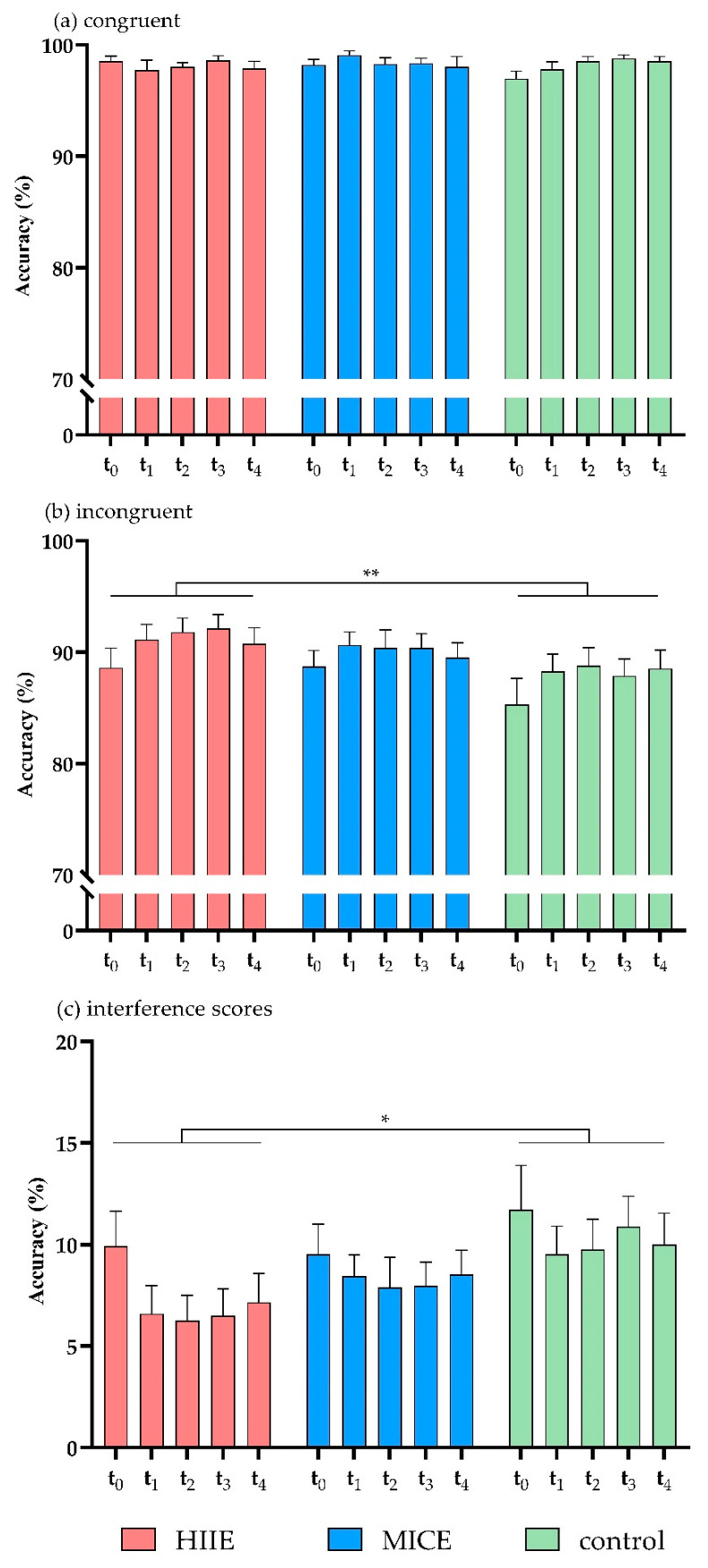
Accuracy for congruent trials (**a**), incongruent trials (**b**), and interference scores (**c**) in the flanker task. HIIE: high-intensity interval exercise; MICE: moderate-intensity continuous exercise. (* *p* < 0.05, ** *p* < 0.01).

**Figure 3 ijerph-18-02687-f003:**
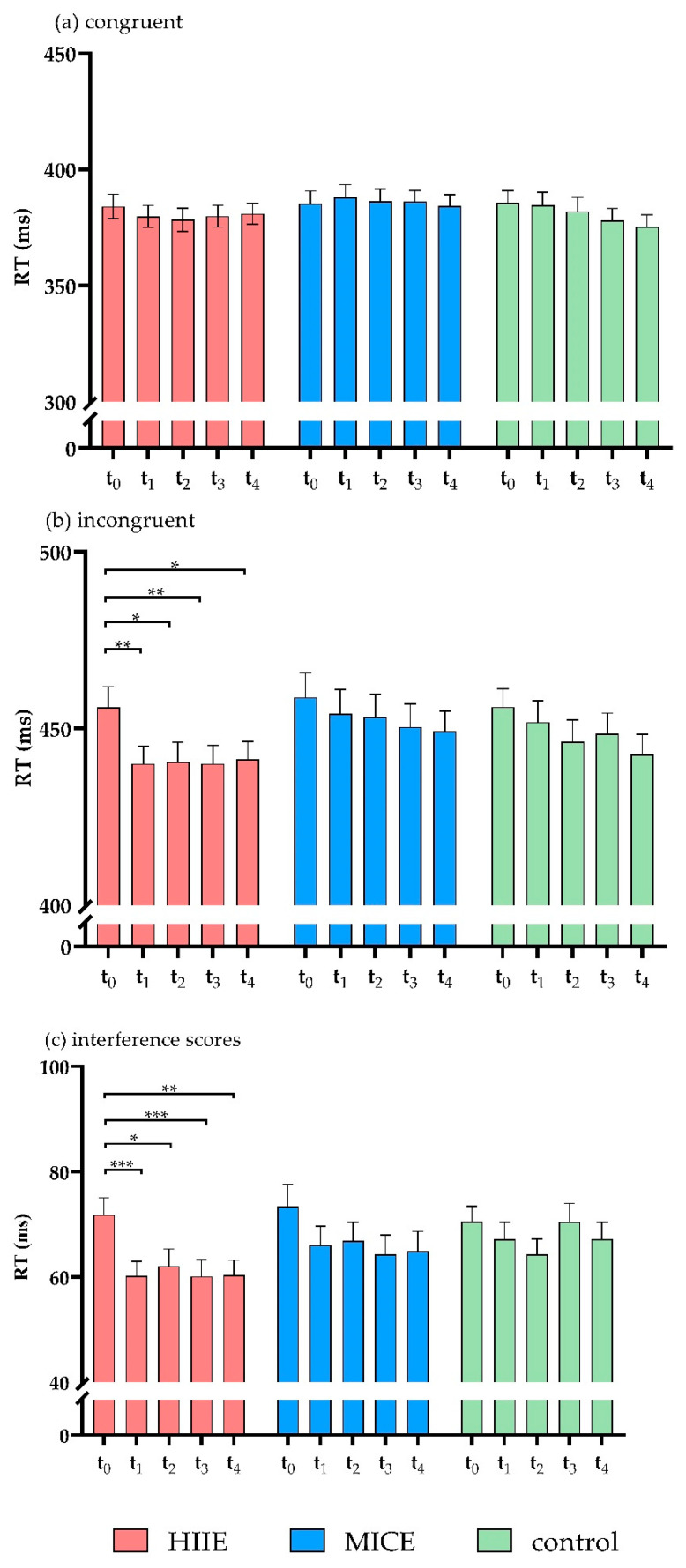
RTs for congruent trials (**a**), incongruent trials (**b**), and interference scores (**c**) of the flanker task. (* *p* < 0.05, ** *p* < 0.01, *** *p* < 0.001).

**Table 1 ijerph-18-02687-t001:** Demographic characteristics data (*M ± SD*).

Variable	*n* = 41
Gender (male/female)	21/20
Age (years)	20.19 ± 1.36
Height (cm)	171.34 ± 9.46
Weight (kg)	62.04 ± 12.18
BMI (kg/m^2^)	20.94 ± 2.62
Estimated HR_max_ (bpm)	193.83 ± 0.95
Mean HIIE HR (bpm)	157.3 ± 8.69
Mean MICE HR (bpm)	134.80 ± 2.85
HIIE RPE	13.41 ± 1.92
MICE RPE	10.85 ± 1.90

Note: BMI= Body Mass Index; RPE = Ratings of Perceived Exertion; HR= Heart Rate.

**Table 2 ijerph-18-02687-t002:** Mean accuracies (%) and Response Times (RTs) (ms) for congruent trials and incongruent trials and interference scores of the flanker task. Data are shown as mean ± standard deviation (*M* ± *SD*).

	Congruent			Incongruent			Interference Score		
	HIIE	MICE	Control	HIIE	MICE	Control	HIIE	MICE	Control
Accuracy t_0_	98.54 ± 2.89	98.21 ± 3.08	96.99 ± 4.27	88.62 ± 11.15	88.70 ± 9.37	85.28 ± 15.09	9.92 ± 10.97	9.51 ± 9.59	11.71 ± 14.09
t_1_	97.72 ± 5.79	99.11 ± 2.24	97.80 ± 4.38	91.14 ± 8.68	90.65 ± 7.39	88.29 ± 9.92	6.59 ± 8.96	8.46 ± 6.67	9.51 ± 8.93
t_2_	98.05 ± 2.47	98.29 ± 3.66	98.54 ± 2.69	91.79 ± 8.13	90.41 ± 10.28	88.78 ± 10.51	6.26 ± 7.89	7.89 ± 9.45	9.76 ± 9.53
t_3_	98.62 ± 2.68	98.37 ± 2.80	98.78 ± 2.08	92.11 ± 8.22	90.41 ± 8.00	87.89 ± 9.74	6.50 ± 8.46	7.97 ± 7.49	10.89 ± 9.52
t_4_	97.89 ± 4.20	98.05 ± 5.82	98.54 ± 2.79	90.73 ± 9.35	89.51 ± 8.52	88.54 ± 10.62	7.15 ± 9.05	8.54 ± 7.60	10.00 ± 9.94
RT									
t_0_	384.10 ± 33.35	385.41 ± 34.00	385.59 ± 33.15	455.94 ± 38.95	458.82 ± 44.94	456.10 ± 32.58	71.84 ± 20.71	73.41 ± 27.38	70.51 ± 18.69
t_1_	379.81 ± 30.01	388.11 ± 34.31	384.64 ± 35.64	440.03 ± 31.86	454.15 ± 43.72	451.82 ± 38.85	60.22 ± 17.54	66.04 ± 23.17	67.18 ± 20.84
t_2_	378.39 ± 32.10	386.29 ± 34.12	381.98 ± 38.60	440.51 ± 36.19	453.19 ± 41.43	446.28 ± 39.52	62.12 ± 20.50	66.90 ± 22.47	64.31 ± 18.86
t_3_	379.90 ± 29.79	386.19 ± 30.60	378.08 ± 32.59	440.03 ± 33.54	450.47 ± 41.62	448.55 ± 37.09	60.13 ± 20.32	64.19 ± 23.68	70.47 ± 22.62
t_4_	380.98 ± 28.98	384.32 ± 31.33	375.42 ± 32.23	441.31 ± 32.25	449.25 ± 36.51	442.63 ± 36.73	60.32 ± 18.43	64.93 ± 23.80	67.21 ± 20.47

## Data Availability

The data presented in this study are available on request from the corresponding author.

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
