# Peer review of "Sustained Effects of High-Intensity Interval Exercise and Moderate-Intensity Continuous Exercise on Inhibitory Control"

_ijerph, 2021, doi:10.3390/ijerph18052687_

Round 1

Reviewer 1 Report

This study examined the immediate and sustained effects of HIIE and MICE on inhibitory control in young adults. Overall, I think the paper is well written, the protocol was sufficiently described. I have some comments for the authors to consider.

Did the authors check the assumptions for using a three-way repeated measures ANOVA, such as the distribution of dependent variables?

The authors should discuss the limitations of the study.

Reviewer 2 Report

Thank you for the opportunity to review the manuscript titled "Sustained effects of high-intensity interval exercise and moderate intensity continuous exercise on inhibitory control". This manuscript examined the immediate and sustained effects of HIIE and MICE on inhibitory control (assessed via the flanker task) in young adults. The results show that HIIE shortened RT immediately following and for up to 90-min post exercise.

This manuscript is clear and concise, and the authors show a revealing command of future studies. The authors have done a great job of providing an informative and meaningful addition to the current study field.

There are only a few minor suggestions I have for the authors. 

Abstract

In the abstract, is it possible that you mention MICE and how it did not effect RT? It seems like it only focuses on the positive result of HIIE on RT and there is no mention of MICE.

Introduction - very nice structure and flow, straight forward and to the point.

Consider adding "the" prior to "intervention" in lines 57-59 - "The flanker task was performed to assess inhibitory control before intervention (t0), and at four time points after intervention, including immediately (t1), 30 min (t2), 60 min (t3) and 90 min (t4)."

Consider revising line 59 for clarity - "Based on the several published literature [20,26]"

Methods 

Is it possible to change the wording on "fitness data" to something more scientific? Or else maybe delete? 

It surprises me the RPE was only 13 and 10 for HIIE & MICE, respectively. According to the ACSM guidelines high-intensity exercise and moderate-intensity exercise would fall into 14-17 and 12-13, respectively. Do you have any idea why this was the case since the HRs were 157 bpm and 135 bpm? The HR max in table is estimated HR max, correct? I think you should clarify this in the table. I'm also not sure you need the sample size row in table 1, this could be included in the table 1 heading or where "Data (M+SD)" is now? (You could then move "Data (M+SD)" to the heading)

Also, why did you not perform a maximal exercise test as a first trial to measure maximal HR so that you could then prescribe a (perhaps) more accurate intensity?

Discussion/Conclusion

Are there any other reasons for the lack of change following the MICE bout? I feel that paragraph in the discussion may need a little more bulk.

I think the second to last discussion paragraph should maybe begin with something related to the mechanisms of such exercise-induced enhancement of inhibitory control (or something along those lines)....

As regards PFC oxygenation, it may also be the case that following MICE there is an increase in PFC O2Hb but no change in inhibitory control in healthy subjects. 

It would have been nice to measure blood lactate also. 

Line 278 - "limitations" or "a limitation". No other limitations here? Target population? No mechanistic data collected?

In your conclusion is it possible you can give an example of a real-life situation whereby the public might be able to use HIIE to improve inhibitory control performance?
